# Sustainable and Printable Nanocellulose-Based Ionogels as Gel Polymer Electrolytes for Supercapacitors

**DOI:** 10.3390/nano12020273

**Published:** 2022-01-15

**Authors:** Rosa M. González-Gil, Mateu Borràs, Aiman Chbani, Tiffany Abitbol, Andreas Fall, Christian Aulin, Christophe Aucher, Sandra Martínez-Crespiera

**Affiliations:** 1Applied Chemistry and Materials, ARTS Department, Leitat Technological Center, C/Pallars, 186-179, 08005 Barcelona, Spain; rosamaria.gonzalez@icn2.cat; 2Novel Energy-Oriented Materials Group, Catalan Institute of Nanoscience and Nanotechnology, ICN2 (CSIC-BIST), Edifici ICN2, Campus UAB, 08193 Barcelona, Spain; 3Energy and Engineering, ARTS Department, Leitat Technological Center, C/de la Innovació, 2, 08225 Barcelona, Spain; mateu@bioo.tech (M.B.); achbani@leitat.org (A.C.); 4Arkyne Technologies SL (Bioo), C/de la Tecnología, 17, 08840 Barcelona, Spain; 5Bioeconomy and Health, RISE Research Institutes of Sweden, Drottning Kristinas väg 61, 114 28 Stock-holm, Sweden; tiffany.abitbol@ri.se (T.A.); andreas.fall@ri.se (A.F.); Christian.aulin@Holmen.com (C.A.); 6Holmen Iggesund, 825 80 Iggesund, Sweden

**Keywords:** nanocellulose, ionic liquid, ionogel, gel polymer electrolyte, renewable energy storage, supercapacitors

## Abstract

A new gel polymer electrolyte (GPE) based supercapacitor with an ionic conductivity up to 0.32–0.94 mS cm^−2^ has been synthesized from a mixture of an ionic liquid (IL) with nanocellulose (NC). The new NC-ionogel was prepared by combining the IL 1-ethyl-3-methylimidazolium dimethyl phosphate (EMIMP) with carboxymethylated cellulose nanofibers (CNFc) at different ratios (CNFc ratio from 1 to 4). The addition of CNFc improved the ionogel properties to become easily printable onto the electrode surface. The new GPE based supercapacitor cell showed good electrochemical performance with specific capacitance of 160 F g^−1^ and an equivalent series resistance (ESR) of 10.2 Ω cm^−2^ at a current density of 1 mA cm^−2^. The accessibility to the full capacitance of the device is demonstrated after the addition of CNFc in EMIMP compared to the pristine EMIMP (99 F g^−1^ and 14.7 Ω cm^−2^).

## 1. Introduction

From all the different types of energy storage systems (ESS), the interest in supercapacitors (SCs) has grown thanks to their energy storage mechanism by electrical double layer, which increases cycle life and allows the use of safer ionic conductors and electrode materials than in traditional batteries [1]. Their promising properties such as high-power density, fast charge–discharge rate and long-cycle stability, have turned SCs into a potentially useful tool for application in flexible and wearable electronics [2,3,4]. In this sense, the use of these quasi-solid electrolytes could play an important role in the production of printed supercapacitors providing flexibility, transparency, stability, and safety to the devices [5]. 

Different types of electrolytes can be used for ESS, including liquid, solid state electrolytes (SSEs) [6] and gel polymer electrolytes (GPEs). GPEs consist basically of a liquid electrolyte, that can be an ionic liquid (IL), or an ionic salt trapped inside a three-dimensional (3D) polymer network [7,8]. ILs, generally a room temperature molten salt, are usually used in GPEs to improve the properties of the ESSs, because of their great electrochemical stability over a wide electrochemical window with minimum decomposition, high ionic conductivity, non-volatility, non-flammability, and good thermal and chemical stabilities [9,10]. Moreover, ionogel electrolytes (IL based gels) present interesting advantages against liquid leakage and corrosion problems, increasing electrochemical stability, and facilitating packaging and manufacturing processes, due to both their liquid-like ionic conductivity and solid-like structural and mechanical features with great flexibility [10].

Recently, in response to an increased demand concerning the sustainable and cost-effective development of electrochemical energy storage devices, great efforts are being devoted to replacing the existing fossil energy resources by new ESSs based on green sustainable materials. As a class of green materials, nanocellulose (NC) has received extensive attention due to its natural abundance, recyclability, facile and diverse modification, large specific area, and dimensional stability [11,12]. NC can be classified into the following categories depending on the source, preparation methods and fiber morphology: (1) cellulose nanofibers (CNF), which refers to the nanoscale cellulose fibrils obtained by fibrillating cellulose under the action of high pressure or mechanical forces, (2) cellulose nanocrystals (CNC) obtained by acid hydrolysis conditions to form nano-sized, rod shaped particles with high crystallinity, and (3) bacterial nanocellulose (BC) with a high aspect ratio obtained by a microbial fermentation process [12,13]. Compared with other biomass-derived green materials, such as lignin, chitin, etc., NC shows great advantages in the field of ESSs [14,15,16,17,18,19,20]: versatile surface modification via functionalization of its hydroxyl groups that allow the integration of other active materials; great thermal stability and extraordinary mechanical properties that can improve the safety properties of the ESSs, through formation of the previous mentioned gel polymer electrolytes (GPEs) [20].

Despite the NC advantages described above, the processing of cellulose-based materials for practical application in energy storage fields, like GPEs, is limited due to their highly ordered structure and crystallinity. However, nanocellulose can be solubilized in some ionic liquids by the disruption of the hydrogen bonding network and the formation of new strong hydrogen bonds with the IL anions [21]. Hence, ILs anions play a predominant role during this process, different than cations with weaker interactions (van der Waals and/or electrostatic interactions) [21,22]. In this way, the use of NC or other sources of cellulose as polymeric matrix for ionogel preparation has been studied.to develop green and biofriendly electrolyte alternatives to fluorinated polymers. Thiemann et al., synthesized an ionic liquid GPE by gelation of microcellulose thin films with 1-ethyl-3-methylimidazolium methylphosphonate [23]. These cellulose ionogels showed high specific capacitances (5–15 μF·cm^−2^), high transparency and flexibility. In another example, a cellulose-based dual network ionogel electrolyte was synthesized by Rana et al. by dissolving phosphorylated microcrystalline cellulose in 1,3-dimethylimidazolium methyl phosphate [DMIM][MeO(H)PO_3_] followed by post-polymerization. The as-synthesized ionogel electrolyte exhibited a high ionic conductivity (2.6–22.4 mS·cm^−1^) over a wide temperature range (up to 120 °C) [22]. 

In this work, we have worked with 1-ethyl-3-methylimidazolium dimethyl phosphate (EMIMP) as the ionic liquid because of the well-known ability of phosphates, among other anions, to dissolve carbohydrates [21,22]. While ionogels from natural cellulose or microcellulose have already been obtained easily [21,22,23,24], we selected carboxymethylated cellulose nanofibers (CNFc) for our studies due to its superior properties: better mechanical performance, and processability; together with the presence of carboxylic groups on the surface, which can boost the strong inter- and intra- molecular interactions inside the resulting ionogel [25]. We present the combination of the EMIMP with CNFc as hosting renewable polymer to provide an easy-to-prepare transparent ionogel electrolyte with exceptional thermal and electrochemical capabilities. For that purpose, four different ratios of CNFc to IL were studied and tested in a coin-cell supercapacitor device as electrolyte (CNFc ratio from 1 to 4) to observe the change in electrochemical performance. Physicochemical characterizations (composition, morphology, rheology, and thermal stability) and electrochemical characterizations (capacity, ionic conductivity and cyclability) were performed to investigate its properties and understand the functionality of the novel nanocellulose-based ionogel electrolytes. 

## 2. Materials and Methods

### 2.1. Materials 

Carboxymethylated cellulose nanofibers were produced by RISE Research Institutes of Sweden from a commercial sulfite softwood pulp (Domsjö Dissolving plus; Domsjö Fabriker AB, Domsjö, Sweden). Ethanol (Rectapur^®^) was purchased from VWR (VWR AB International, Spånga, Sweden). Monochloroacetic acid (99%, ACS reagent, ClCH_2_COOH), acetic acid (ACS reagent), iso-propanol (ACS reagent), sodium hydroxide (ACS reagent), sodium hydrogen carbonate (ACS reagent) and methanol (ACS reagent) were purchased from Sigma Aldrich (Stockholm, Sweden). 1-ethyl-3-methylimidazolium dimethyl phosphate (>98% grade) (EMIMP) was purchased at Iolitec (Heilbronn, Germany) and used as received. Active Carbon YP50F (Kuraray, Kurashiki, Japan), conductive carbon TIMCAL C 65 (Cabot), 1-Methyl-2-pyrrolidone (NMP, Scharlau, Sant Feliu del Llobregat, Spain) and 5130 PVDF Solef (Solvay, Barcelona, Spain) were used in the formulation of the active layers of the supercapacitors. The fabricated electrodes were assembled in CR2032 coin cells from MTI by using cellulose separator Kodoshi (Nippon Kodoshi Corporation, Kochi, Japan) and 1-ethyl-3-methylimidazolium dimethyl phosphate (EMIMP, >98%, Iolitec) or EMIMP:CNFc mixtures (96:4 to 99:1 in ratio) as electrolyte. 5 samples electrolytes are reported including EMIMP pure electrolyte and 4 mixtures adding 1%, 2%, 3% and 4% of CNF in EMIMP ionic liquid. It has not been possible to produced jelly electrolyte with higher rate of CNF due to the loss of the mechanical properties not allowing to print and process the component. The preparation of the electrode is identic for each symmetric coin cell and described in Section 2.3. Each electrode is prepared by coating of the jelly electrolyte directly on the top of the dried carbon electrode and then the CR2032 coin cell is assembled with staking 2 symmetric electrode separated by Kodoshi nanocellulose separator (thickness 50 µm) without further addition of electrolyte or solvent.

### 2.2. Carboxymethylated Cellulose Nanofibers (CNFc) Fabrication

CNFc fabrication was done according to the general methodology described in the literature [26]. Most commonly, CNFc is prepared with a total charge of ~600 µeq/g, corresponding to a carboxmethyl degree of substitution (DS) of 0.1 [27]. In this work, a higher charge DS 0.3 CNFc was produced as has been recently described [28]. Generally, resulted pulp is solvent exchanged to ethanol followed by carboxymethylation reaction, which begins by impregnating the pulp with a solution of monochloric acetic acid in isopropanol, followed by transfer of the pulp to a heated methanolic solution of NaOH in isopropanol and reaction under reflux for 1 h. Compared to the reaction conditions for DS 0.1, to achieve DS 0.3, the dosage of monochloroacetic acid is increased 4.4-fold and NaOH 2.7-fold. After reaction, the carboxymethylated pulp is washed with NaHCO_3_ to neutralize, and finally washed with water to remove excess reagents, until the wash has a conductivity below 10 µS/cm. To generate CNFc, the carboxymethylated pulp is microfluidized (Microfluidizer M-110EH, Microfluidics Corp., Newton, MA, USA) at 1700 bar (1 pass) to give a suspension of DS 0.3 CNFc at 1 wt%. The CNFc was then subjected to lyophilization and morturation before use in Leitat (Barcelona, Spain). 

### 2.3. CNFc Based Ionogel Preparation 

In Figure 1 it is shown a scheme about the NC-ionogel preparation process. For this, freeze-dried CNFc from an aqueous dispersion (at 3 wt%) was mechanically mixed with different ratios of EMIMP (p/p, EMIMP/CNFc: 99:1, 98:2, 97:3 and 96:4, referred as EMIMP/CNFc 99:1, EMIMP/CNFc 98:2, EMIMP/CNFc 97:3 and EMIMP/CNFc 96:4) with heating (100 °C) for 16 h to obtain a yellowish viscous gel (final appearance pictures are depicted in Appendix A). The as prepared ionogel could be used without any additional step in ambient air, showing long-term stability since no destabilization phenomena, like phase-separation, have been observed after approx. 2 years.

### 2.4. Supercapacitor Preparation 

The active ink used in the supercapacitor application was fabricated by mixing NMP, active carbon YP50F, conductive carbon TiMCALC65 and binder PVDF at dry weight ratio of 80:10:10 by using a Dipermat (VMA-GETZMANN GmbH, Reischsof, Germany) with a 3D printed head reproducing the shape of an industrial planetary mixer but adapted for lower volume of ink. The prepared ink was printed onto a 20 µm aluminum foil by bare coating (gap of 100 µm) and dried for 2 h at 120 °C in vacuum obtaining a dry coating of 50 µm thickness and active material loading of 1.8 mg cm^−2^. For the CNFc-ionogel printing, heating to 80–100 °C allowed direct printing using a Doctor Blade technique (gap of 150 µm bar) onto a carbon electrode surface as shown in Figure 2.

### 2.5. Characterization

X-ray diffraction (XRD) patterns were recorded for each sample on a diffractometer (Malvern PANalytical X’pert Pro MPD, X-ray Diffraction Facility, *Institut Català de Nanociència i Nanotecnologia*) at room temperature from 10 to 50 2Ɵ. Fourier transform infrared spectroscopy were recorded with a Shimadzu Iraffinity-1S spectrometer in ATR mode. The specimens were measured directly with a scan range from 400 to 4000 cm^−1^. The thermogravimetric analysis (TGA) of the samples (5–10 mg) were carried out by using a TGA/DSC (Mettler-Toledo, University of Alicante, Alicante, Spain), equipped with autosampler and gas-controlled unit, from room temperature to 900 °C with heating rate of 20 °C/min under N_2_ conditions. Scanning Electron Microscopy (SEM) were performed in a Zeiss Merlin FE-SEM (Carl Zeiss Microscopy, Autonomous University of Barcelona, Bellaterra, Spain) with samples mounted directly on SEM stubs with adhesive carbon tape and metalized with Au thin film (E5000 Sputter Coater. Polaroid Equipment Limited. Watford). Images were recorded at different steps of magnification. The rheologic properties of EMIMP and all fabricated electrolytes were analyzed by using the rheometer Bohlin CVO 100–901 in the share rate range from 0.0517 to 888 s^−1^.

The electrochemical testing was performed with a multichannel potentiostat VMP3 (Bio-Logic) run with the EC-Lab software. Cyclic voltammetry was performed between 0 < V < 2.0, and with a scan rate between 10 < mV s^−1^ < 50, to investigate side reactions, evaluate the potential window of stability and the device behavior. Galvanostatic measurements at a constant areal capacity of mA·cm^−2^ and between 0 < V < 2.0 V) were used to calculate the specific capacity (mAh·g^−1^ of AC, energy (unit), power (unit), and equivalent series resistance (ESR) is calculated from the ohmic drop between of the developed supercapacitors. Electrochemical impedance spectrum measurements were carried out with frequency range of 50 mHz to 1 MHz. All these analyses were carried out by VMP3 from Biologic. Finally, a long cycling at 1 mA·s^−1^ for 10,000 cycles were carried out by Neware cycler. The electrochemical CR2032 coin-cell type used for the tests is built from the stack of two symmetric electrodes separated by cellulose membrane separators and with 40 µL of EMIMP or 150 µm (wet) bar-coated EMIMP:CNFc gel as electrolyte ensuring it excess.

The cell capacitance (*C_cell_)* of a symmetrical capacitor can be calculated according to Equation (1) [29]:
(1)$$C_{cell}=\frac{C_{e}}{2};C_{e}=2C_{cell}$$
where *C_e_ = C_+_ = C_−_* are the specific capacitance of the two electrodes in mAh·g^−1^ of activated carbon, and the cell capacitance of a supercapacitor can be calculated from the charge-discharge experiment Equation (2):(2)$$C_{cell}=\frac{I}{mdV/dt}$$

This can be then used, when combined with *C_cell_* equation to obtain the electrode capacity, and subsequently obtain information on the capacity of an investigated material Equation (3):(3)$$C_{e}=\frac{4I}{mdV/dt}$$
where *I* (A) is the discharge current, *m* (g) the total mass of active materials of two electrodes and *dV/dt* the slope of the discharge curve between 80% and 40% of the cut-off voltage (*V*). From the total specific cell capacitance of two-electrode system (*C_cell_*), the maximum energy (*E_max._*) and maximum power (*P_max._*) of a supercapacitor cell can be calculated according to Equations (4) and (5):(4)$$E_{max.}=\frac{1}{2}C_{cell}V^{2}\;$$
(5)$$P_{max.}=\frac{V^{2}}{4ESR}$$
where, *V* (V) is the cell voltage minus the ohmic drop, and *ESR* (Ω) is the total equivalent series resistance of the supercapacitor. From the EIS, the ionic conductivity of the different gels has been calculated using the following formula Equation (6):(6)$$\sigma =\frac{l}{R_{1}A}$$
where σ is the ionic conductivity (S m^−1^), *l* (m) is the thickness of the membrane, *A* (m^2^) is the contact area between the electrode and the electrolyte and *R_1_* (Ω) is the ionic resistance at high frequency corresponding to the first interception of the circle in the Nyquist plots.

## 3. Results and Discussion

An extensive characterization of the prepared CNFc-ionogels have been realized to study their composition, homogeneity, thermal and rheological properties. XRD patterns for each IL/CNFc ratio, pristine CNFc (lyophilized) and EMIMP are shown in Figure 3a. Broad signals are obtained in all cases, obtaining major intensities at 12° and 25° for all the EMIMP/CNFc mixtures, which are related to the ionic liquid EMIMP (triangle signals), with a minor signal present as a broad band at 15° corresponding to (002) nanocellulose crystalline plane (star signals). The major component of ionic liquid ensures a good ionic conductivity and a good gel flexibility due to their mainly amorphous nature. Fourier transform infrared spectroscopy (FTIR) of each CNFc-ionogel were also recorded and are depicted in Figure 3b, where the IR profiles of pure CNFc and EMIMP are also added. As in case of previous XRD studies, the EMIMP has the major intensity, showing its typical bands of -CH from the imidazolium ring (3060 and 1570 cm^−1^), the -CH_2_ st and -CH_3_ st (2920–2970 cm^−1^), -CH_3_ bend (1450 cm^−1^) and -CH_2_ bend (780 cm^−1^) from the ethyl chains, the -C-N st at 1240 cm^−1^ and finally, the -P-O-C (930–1050 cm^−1^) and -P=O (1090–1200 cm^−1^) vibrations of the diethylphosphate group. CNFc (highlighted in blue) shows an -OH band at 3400 cm^−1^, which increases with the higher ratio of CNFc in the mixtures, a -C=O peak related with the carboxylated groups at 1600 cm^−1^, which appears slightly shifted to 1650 cm^−1^ for all ionogel mixtures, and a -C-O-C streaching at 1000 cm^−1^ from CNFc glucose units, which can not be appreciated in the mixtures due to overlaping with EMIMP P=O and P-O-C signals.

Thermal properties (TGA) were studied for each CNFc-ionogel and are depicted in Figure 3c. The TGA profiles shown in all cases a two-step weight loss at temperatures below 100 °C and above 250 °C, which can be reasonably attributed to evaporation of adsorbed water (less than a 15% wt) and the degradation of CNFc and the EMIMP, respectively. This data shows the high thermal stability of the new gel electrolyte. DSC experiments have been also studied to determine the degradation temperatures for each ionogel in comparison to pure EMIMP (Appendix A), obtaining similar results between them (290 °C for EMIMP, to 293 °C for EMIMP/CNFc 96:4). The addition of smalls amounts of nanocellulose has a minor affection on the decomposition temperatures, improving them slightly. ILs are mostly Newtonian fluids and more viscous than most common molecular solvents [30]. Their ionogels show similar viscoelastic properties under stress, providing a higher-quality electrode/electrolyte contact, in comparison with other rigid solid-state electrolytes [10,31]. In the case of this work, viscosity measurements (Appendix A) reveal that almost all the prepared NC-ionogels have similar viscosity characteristics, except for the mixture EMIMP/CNFc 96:4, that reveals a thixotropic behavior, typical of NC. Regarding the apparent viscosity dependence of the applied shear stress (Figure 3d), all NC-ionogels, and CNFc (1% wt, gel), show a reduction in the viscosity, which is more pronounced in the ionogels with a higher CNFc content (EMIMP/CNFc 96:4 and EMIMP/CNFc 97:3), with no changes in case of pure EMIMP. This characteristic, named shear-thinning [32], is common of pseudoplastic fluids and consents on a more effective particles flow through the liquid phase, reducing the viscosity of the solution. It is intrinsic of the nanocellulose and its derivatives [33,34], and it is responsible for the good printability of the presented ionogels, while the desired mechanical properties were maintained.

High resolution scanning electronic microscopy (FE-SEM) was done for the ionogel EMIMP/CNFc 96:4 to study its morphology, showing a smooth and homogeneous surface. Moreover, an EDS mapping of nitrogen, phosphorous and sodium elements that are present in the ionic liquid (N and P) and the nanocellulose (Na) was also performed to confirm the homogeneity of the GPE. (See Figure 4).

In the same way, to use these NC-based ionogels as electrolytes for supercapacitors, their electrochemical performance and properties were first evaluated. Symmetric cells were assembled in CR2032 coin cell to test the electrochemical performance of the EMIMP electrolyte, alone and with the different CNFc ratios, using a cellulosic Kodoshi separator. The cyclic voltammogram at 10 mV·s^−1^ (See Figure 5a) of the supercapacitor with EMIMP electrolyte shows a near rectangle shape, approximated to the ideal situation of double layer capacitor (or supercapacitor). There is no visible redox peak from Faradaic current over the potential region. However, a peak onset appears as 2 V is approached. At high scan rates (See Figure 5b), the shape of the CV curve was distorted due to an increase of the internal resistance and the peak onset near 2 V was shifted to higher voltage, due to the increase in cell resistance at higher scan rate.

When CNFc is added to the EMIMP ionic liquid, the capacity of the device increases with the addition of CNFc, the peak onset near 2 V in the oxidation zone becomes more evident and a new one between 0 and 0.5 V in the reduction zone appears (Figure 5a). At high scan rates (Figure 5b), the shape of the CV curve was distorted due to an increase of the internal resistance but to lesser extent than with EMIMP alone and the peaks onset near 2 V and between 0 and 0.5 V disappear. However, no clear differences in capacity and resistance between the electrolytes with different ratios of EMIMP/CNFc can be appreciated from the cyclic voltammetry studies.

In order to clarify if the peak onset observed in the cyclic voltammetry was caused by the electrolyte degradation or because of its own behavior (as has been seen before in different IL electrolytes) [35], a floating test was carried out [36]. The supercapacitors with the different studied electrolytes were charged at 2V in a stepwise ramp and the voltage was maintained for a period of 24 h, as shown Appendix A. The leakage current obtained at each time was recorded and could be used to directly assess the importance of the electrochemical degradation of the electrolytes in the fully charged state. The results showed that the leakage current of the EMIMP alone was 6 times higher than the one obtained by adding CNFc (i.e., 120 µA vs 20 µA). The data obtained for the different EMIMP/CNFc electrolytes were low enough to determine that the peak onset showed in the cyclic voltammogram was not caused by electrolyte degradation [37]. Moreover, these results demonstrated that the addition of CNFc increase the EMIMP voltage resistance. The capacity and resistance of the supercapacitors with EMIMP and with the gel electrolytes (by the introduction of different rates of CNFc in the EMIMP) were evaluated by galvanostatic charge-discharge tests. All these parameters were calculated using the equations described in Section 2 and the results are showed in Appendix A and Figure 6. In the case of capacities calculated from cyclic voltammetry, as shown Appendix A, the capacitance tendency in all cases is decreasing at higher scan rate. Higher specific capacity is measured after addition of CNFc, which is online with galvanostatic charge-discharge results. The supercapacitors with CNFc in the electrolyte exhibit a smaller ESR in all intensity ranges than the supercapacitors with only EMIMP (4.0–10.2 Ω for different rates of EMIMP:CNFc and 14.7 Ω for EMIMP alone). These results also indicate that EMIMP/CNFc gel electrolytes (in all the studied ranges) have higher ionic conductivity than EMIMP alone, and this has been confirmed by the electrochemical impedance spectroscopy showed in Figure 7. In addition, according to Equations (2) and (3), the capacity for the electrolyte EMIMP and EMIMP/CNFc with different ratios were calculated to be 99 and 132–160 F g^−1^ of YP50F active material at current density of 1 mA cm^−2^. The introduction of CNF in the electrolyte significantly increases the capacity of the electrode, and this increase on capacity and decrease on ESR is also significant between the four gel electrolytes with different CNFc ratios. Maximum of capacity is reached for EMIMP/CNFc 97:3 ionogel but presenting similar capacity than the EMIMP/CNFc 96:4 ratio. However, the ESR of EMIMP/CNFc 96:4 is clearly lower than the other gel electrolytes that present similar values. In addition, we previously demonstrated that this increase on capacity does not come from the electrolyte (or CNFc) degradation by the floating test, rather this increment on capacity suggests that part of the CNFc present in the electrolyte is displaced to the electrode surface (due to his high polarity), creating an extra porous structure where more ions are allocated during the charge process, and therefore, increasing the capacity of the device. It has not been tested above 96:4 due to the poor mechanical of gel above this range.

The capacity variation with the cycles is shown in Appendix A**.** The supercapacitors with EMIMP show a good capacity retention of 80% after 10,000 cycles at 1 mA cm^−2^ between 0 < V <2. The supercapacitors with CNFc had a quick capacity decay in the first 2000 cycles (~0.020 mAh g^−1^ cycle^−1^ of capacity decay) and then present a moderate degradation (~0.001 mAh g^−1^ cycle^−1^ of capacity decay) reaching the 10,000 cycles with a capacity retention below 50%. It is possible that during the cycles, the CNFc cannot be retained in the matrix of the electrolyte and that creates a larger SEI in the first cycle, starting to cover part of the carbon porous and therefore limiting the capacity of the electrodes.

To rule out a possible dissolution of CNFc to carboxymethyl cellulose (CMC) units, an ionogel with a 96:4 ratio EMIMP/CMC was prepared and assessed. Hence, in the case of use CMC instead CNFc, the electrochemical performance of these supercapacitors, shown in Appendix A, provides specific capacitance values of 123 F g^−1^, being like EMIMP/CNFc 99:1 ionogel values. Moreover, it has been observed the formation of a solid precipitates by mixing CMC to EMIMP, presumably due to a lack of interaction between the CMC and IL because of the diminution of available -OH groups. As it has been mentioned, the presence of these hydroxyl groups plays an important role in the dissolution of cellulose in ILs, due to they are the responsible of the formation of new hydrogen bonds between the NC and the ILs anions [38,39].

Finally, electrochemical impedance spectroscopy (EIS) for the supercapacitors with EMIMP/CNFc gel electrolytes and EMIMP are shown in Figure 7a. The equivalent circuit was made with EC-Lab (Figure 7b). The Nyquist plots show a semi-circle at high frequencies region and a line perpendicular to the X-axis at low frequencies that correspond the ionic diffusion and the double layer formation at the surface of the conductive solid phase. R1 is mostly corresponding to the ionic conductivity in the dielectric, both gel and membrane separator. R2 is referring to the charge transfer both ionic and electronic in the bulk of the electrode. Q1 and Q2 are constant phase element simulating the real behaviour of a pure capacitor. The first thing to be considered is a charge transfer resistance (R_ct,_ that includes the ionic resistance), which could be intensively investigated in the semi-circle loop appearing in the middle frequency region. A decrease in resistances is shown in supercapacitors with CNFc compared to supercapacitors with only EMIMP. This decrease in resistances has been observed above all in the value of the charge transfer resistance (R_ct_). These results are in line with the ones obtained in the CV and galvanostatic tests. The real and imaginary capacitance versus the frequency is given in supporting in Appendix A and using the definition from P.L. Taberna and et al. [40]. The real part of the capacitance shows the variation of the available stored energy with the frequency. The imaginary part of the capacitance represents the losses that occur during charge storage. In these graphs, can observe that the maximum capacitance is reached faster at higher frequencies in the case of the blended electrolyte containing CNFc versus EMIMP electrolyte. No real difference is observed within the different percentage of CNF used in this studied. No degradation of the electrolyte is demonstrated by the floating test, also the same active material and electrode systems have been used for all electrolyte testing, then we assume no intrinsic supplementary capacity is reached by adding CNFc to EMIMP but a faster access to this maximal capacity is demonstrated.

All the values of ionic conductivity (σ, mS cm^−1^) and resistances (R1 and R2, Ω) are included in Table 1. The increase in conductivity with the addition of CNFc was not expected because of the important increment on the electrolyte viscosity (almost 0 to 60 Pa ·s), Figure 3d and Appendix A). However, as can be seen in Table 1, adding nanocellulose has helped to increase the ionic conductivity of the GPEs, causing a decrease in the internal resistance of the system (0.32–0.94 mS cm^−1^ for EMIMP/CNFc ionogels against 0.26 mS cm^−1^ for EMIMP). Plenty of polar functional groups in the system helped to increase the dielectric constant of the gel electrolytes, which is proportional to the capacity assuming all electrode surface area and distances between the two electrodes are the same for each cell assembled. This also increases the interaction between ions and polymer chains and helps to dissociate these ion pairs into separate ions. As a result, more free ions are generated and consequently, the ionic conductivity is enhanced. Moreover, the abundant polar functional groups supplied by cellulose offer an ionic conduction channel by aiding facile ion hopping onto nearby polar groups However no tendency could be measured with the increased ratio of CNFc.

## 4. Conclusions

Novel GPEs have been prepared by the addition of different CNFc ratios (from 1 to 4) to EMIMP, showing excellent thermal and mechanical properties. Contrary to the usual liquid electrolytes, these new NC based GPEs not only improve safety issues, but also show a good printability, very interesting for printed electronics applications. Moreover, enhanced electrochemical properties have also been measured. In case of specific capacitance, EMIMP/CNFc ionogels have shown a significant increase (160 F g^−1^) compared to the pristine IL (99 F g^−1^), and its internal resistances have considerable decreased from 14.7 Ω cm^−2^ to 10.2 Ω cm^−2^ at a current density of 1 mA cm^−2^. Although the viscosity (Figure 3d) of the electrolyte increases after CNF addition, which should promote lower ionic conductivity, all electrochemical measurements show both higher ionic conductivity and capacity with CNF addition. These data confirm that the addition of polar groups coming from nanocellulose play a beneficial role. However, one of the points to be improved in future experiments is to maintain the capacitance over the cycles (50 % decrease in capacitance in the first 2000 cycles). A drastic loss of the capacity is observed for all samples containing CNF after long cycling (Appendix A), although the leakage current test showed no electrolyte degradation (Appendix A). With these results we assume the addition of CNF is beneficial at device level, reducing the initial internal resistance and increasing the quality of the interface between the electrolyte and the electrode. But this effect disappears within the cycling, where samples with and without CNFc show similar capacity after 2.000 cycles A possible passivation of the electrode that it is reducing both capacity and ionic conductivity is not discarded and needs to be further studied. Moreover, investigations about the use of CNFc vs CMC have been also performed, showing that the use of the nanofibrils of cellulose form (CNFc) provides better solubility in EMIMP and higher electrochemical performances. With all this results we can conclude that these new GPEs have shown great potential for applications in energy storage devices.

## Figures and Tables

**Figure 1 nanomaterials-12-00273-f001:**
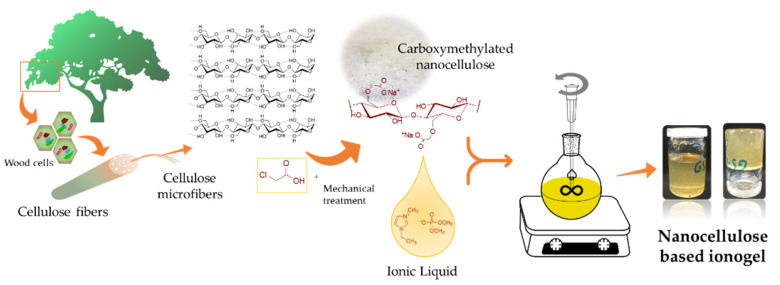
Schematic preparation of CNFc-based ionogels.

**Figure 2 nanomaterials-12-00273-f002:**
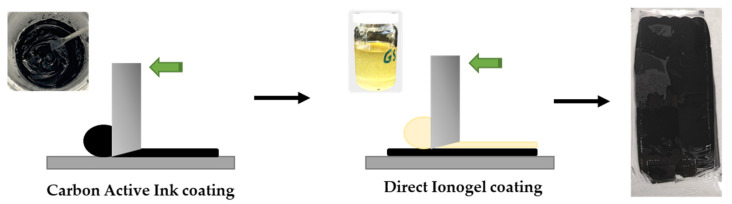
Direct printing process by Doctor Blade technique of the CNFc-ionogel onto the electrodes.

**Figure 3 nanomaterials-12-00273-f003:**
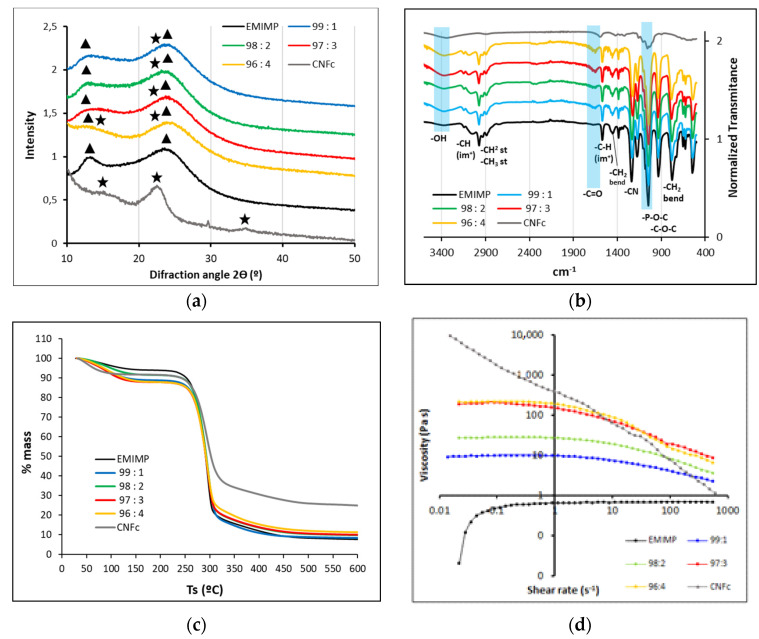
(**a**) XRD; (**b**) FTIR; (**c**) TGA and (**d**) Rheological data from CNFc-EMIMP GPE, including pristine CNFc (gel form at 1%wt) and EMIMP.

**Figure 4 nanomaterials-12-00273-f004:**
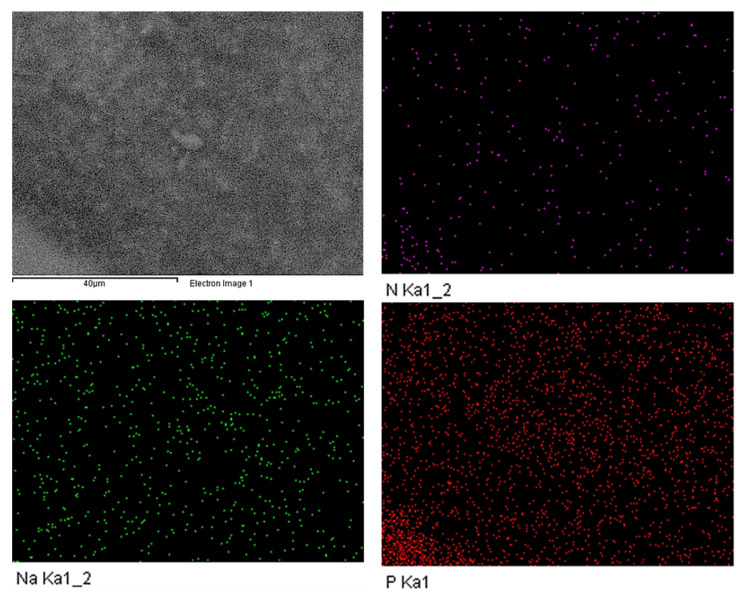
FE-SEM image of EMIMP/CNFc 96:4 gel and its EDS mapping on N, Na and P elements.

**Figure 5 nanomaterials-12-00273-f005:**
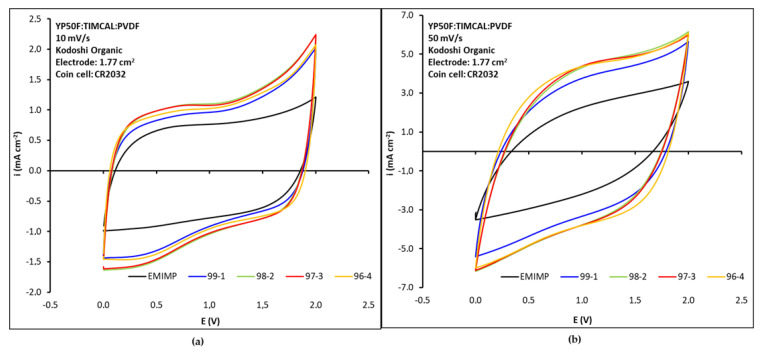
CV (cyclic voltammetry) curves at (**a**) 10 mV s^−1^; (**b**) 50 mV s^−1^ scan rates for each EMIMP/CNFc ionogel (99:1, 98:2, 97:3 and 96:4). Pristine EMIMP is also added for comparison.

**Figure 6 nanomaterials-12-00273-f006:**
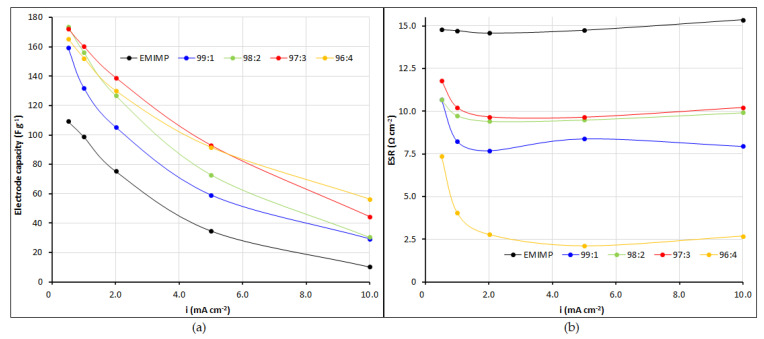
(**a**) Electrode capacity and (**b**) internal resistance (ESR) of the supercapacitors with EMIMP and EMIMP/CNFc mixtures (99:1, 98:2, 97:3 and 96:4) as electrolyte at different charge-discharge current rates.

**Figure 7 nanomaterials-12-00273-f007:**
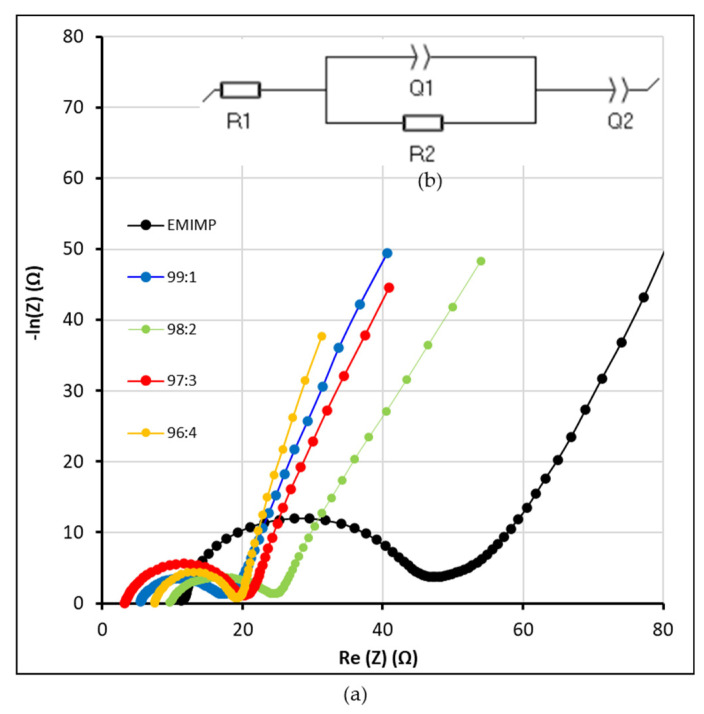
(**a**) Nyquist plots of the different EMIMP/CNFc gel electrolytes and EMIMP ionic liquid; (**b**) Equivalent circuit of the system.

**Table 1 nanomaterials-12-00273-t001:** Ionic conductivity and resistances of the different EMIMP/CNFc gel electrolytes and pristine EMIMP ionic liquid.

Electrolyte	σ (mS cm^−1^)	R1 (Ω)	R2 (Ω)
EMIMP	0.26 ± 0.03	11.14	36.02
EMIMP/CNFc 99:1	0.57 ± 0.03	5.07	12.76
EMIMP/CNFc 98:2	0.32 ± 0.03	8.92	15.77
EMIMP/CNFc 97:3	0.94 ± 0.03	3.08	17.03
EMIMP/CNFc 96:4	0.44 ± 0.03	6.5	13

## Data Availability

The data presented in this study are available on request from the corresponding author.

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
