# Peer review of "Sustainable and Printable Nanocellulose-Based Ionogels as Gel Polymer Electrolytes for Supercapacitors"

_nanomaterials, 2022, doi:10.3390/nano12020273_

Round 1

Reviewer 1 Report

I have attached the comments and suggestions for authors in the attached word file. 

Thank you.

Author Response

I submit the answer

Reviewer 2 Report

This paper reported a nanocellulose-based ionogel as electrolyte for supercapacitors. Carboxymethylated cellulose nanofibers (CNFc) was used to boost the strong inter- and intra- molecular interactions inside the ionogel. The prepared ionogel via an easy method showed good electromechanical properties. Based on the recent research on the gel polymer electrolytes, it is an interesting work to the authors of Nanomaterials. The following comments might be helpful to improve the quality of the manuscript.

1) These might be English errors:

Line 34 “Their promising properties such a high-power density ..” in which such a should be such as

Line 134 “In Figure 1 is shown a scheme about...”

Line 266 “morfology” should be “morphology”

Line 335 “The supercapacitors with EMIMP shown a good capacity..”

Line 336 “after 10.000 cycles..” in which it might be 10,000 rather than 10.000.

Line 393 “but also and due to its …”

Many linking errors appeared, such as line 247, line 254, line 298, line 310, line 334 and so on.

2) In the introduction, more related references should be cited especially in the part of line 51-84, the description of the NC related research on ESSs.

3) In figure 3, what’s the state of CNFc? From XRD to viscosity, what kind of CNFc was used? Suspension in water or Freezing-dried powder/gel? It is critical to understand the viscosity curve of CNFc.

Author Response

I submit the answer

Reviewer 3 Report

The paper reports on preparation of gel polymer electrolytes for supercapacitors using nanocellulose-based ionogels containing 1-ethyl-3-methylimodazolium dimethyl phosphate. Comprehensive experimental results are presented confirming the advanced properties of these novel electrolytes and the enhanced characteristics of the corresponding supercapacitors. The paper is worth of being published in Nanomaterials journal.

I have the following comments:

1) Please provide the photographic images showing the appearance of the ionogels.

2) Please comment on the long-term stability of the ionogels since this is an important characteristic of these type of materials. Are there any signs of phase separation upon long-term storage or exploitation of the ionogels?

3) The discussion of the results should be expanded to include additional comparison of the materials and devices prepared with their existing counterparts.

Author Response

i submit the answer

Reviewer 4 Report

Evidently, supercapacitors of high energy and safety are highly demanded. In this ms it reported the preparation of a gel polymer electrolyte based on 1-ethyl-3-methylimidazolium dime- 21 thyl phosphate (EMIMP) ionic liquid with carboxymethylated cellulose nanofibers (CNFc) and its use in supercapacitors. The results are intresting, and it is recommended to publish in this journal after major revision:

  1. Some important literatures such as Faxing Wang et al. (Adv. Mater., and Chem. Soc. Rev.) and Bin Wang et al. (Energy Mater. , 2021, 1: 100010) should be mentioned and discussed.
  2. The results in Tab. 1 is not consistent with those in Fig. 5. The authors should re-test the ionic conductivity with the change of IL and CNFs. It is clear that there are some problems. 
  3. The data in Fig. 6 are also abnormal since it is against normal sense. It is well-known that the ionic conductivity and rate capability will increase with the content of IL.
  4. If possible, the energy density of this system should be provided including their change with power density. 

Author Response

i submit the answer

Reviewer 5 Report

Q1: The author summarizes “CNFc (highlighted in blue) shows an -OH band at 3400 cm-1, which increases with the higher ratio of CNFc in the mixtures” Why? How does the author rule out the influence of H2O during the test?

Q2: “At high scan rates (See Figure 5b), the shape of the CV curve was distorted due to an increase of the internal resistance and the peak onset near 2 V disappeared.” Please give relative characters and literatures to confirm your view.

Q3: In the experimental part, how to control the thickness of the membrane.

Q4: In the line 17 of page 7, 'microcroscopy ' should be corrected as 'microscopy'. In the line 18 of page 7, 'morfology' should be corrected as 'morphology'. The spelling should be carefully polished in the manuscript.

Q5: “High resolution scanning electronic microcroscopy (FE-SEM) was done for the ionogel EMIMP/CNFc 96:4 to study its surface morfology.” However, the surface morphology is not expounded. Please explain? Meanwhile, the authors should provide high-resolution images in Figure.4.

Q6: “In addition, we previously demonstrated that this increase on capacity does not come from the electrolyte (or CNFc) degradation by the floating test.” The author should cite previous relatively literature to support this conclusion.

Q7: Some expressions and sentences should be polished, and some related literatures (Electrochimica, Acta, 2019, 305: 563-570; Journal of The Electrochemical Society, 2020,167: 020550; doi: 10.1007/s12598-020-01429-x; doi: 10.1007/s12598-020-01678-w; doi: 10.1016/j.jcis.2021.09.067; doi: 10.1007/s12598-020-01526-x) should be referred to modify the manuscript.

Author Response

i submit the answer

Round 2

Reviewer 4 Report

It is recommended to publish.